# Impact of Reducing Statistically Small Population Sampling on Threshold Detection in FBG Optical Sensing

**DOI:** 10.3390/s24072285

**Published:** 2024-04-03

**Authors:** Gabriel Cibira, Ivan Glesk, Jozef Dubovan, Daniel Benedikovič

**Affiliations:** 1Institute of Aurel Stodola, Faculty of Electrical Engineering and Information Technology, University of Zilina, Komenskeho 843, 03101 Liptovsky Mikulas, Slovakia; 2Department of Multimedia and Information-Communication Technologies, Faculty of Electrical Engineering and Information Technology, University of Zilina, Univerzitna 1, 01026 Zilina, Slovakia; ivan.glesk@uniza.sk (I.G.); jozef.dubovan@uniza.sk (J.D.); daniel.benedikovic@uniza.sk (D.B.)

**Keywords:** statistically small population sampling, two-sided sampling, one-sided sampling, threshold detection, fiber Bragg gratings, FBG sensing, optical spectrum analysis

## Abstract

Many techniques have been studied for recovering information from shared media such as optical fiber that carries different types of communication, sensing, and data streaming. This article focuses on a simple method for retrieving the targeted information with the least necessary number of significant samples when using statistical population sampling. Here, the focus is on the statistical denoising and detection of the fiber Bragg grating (FBG) power spectra. The impact of the two-sided and one-sided sliding window technique is investigated. The size of the window is varied up to one-half of the symmetrical FBG power spectra bandwidth. Both, two- and one-sided small population sampling techniques were experimentally investigated. We found that the shorter sliding window delivered less processing latency, which would benefit real-time applications. The calculated detection thresholds were used for in-depth analysis of the data we obtained. It was found that the normality three-sigma rule does not need to be followed when a small population sampling is used. Experimental demonstrations and analyses also showed that novel denoising and statistical threshold detection do not depend on prior knowledge of the probability distribution functions that describe the FBG power spectra peaks and background noise. We have demonstrated that the detection thresholds’ adaptability strongly depends on the mean and standard deviation values of the small population sampling.

## 1. Introduction

Measuring the resonant wavelengths of fiber Bragg grating (FBG) sensors, finding their fingerprints, or classifying FBGs themselves in optical sensing systems require denoising the acquired FBG spectral peaks [1,2,3,4,5,6,7]. Meanwhile, all important properties of FBG spectral peaks have to be preserved.

To mitigate the impact of the background noise on the quality of the optical signals in telecommunications and sensing systems, hardware pre-processing using real-time wavelength filtering has been typically used [8,9,10]. Software approaches based on verification or post-detection algorithms can be leveraged to detect signals in a complex or fluctuating background noise environment [11,12]. In the latter case, a variety of statistical detection techniques are usually applied. Such methods utilize a preset level of an allowable detection threshold *τ*, relying on the signal-to-noise ratio (SNR) evaluation [12,13].

The reviewing article [14] deals with Brillouin scattering methods of determining the frequency shift of signals and approximation methods in fiber optic metrology and distributed optical fiber sensing. Based on available filtering, fitting, approximation, correlation, and control techniques, they aim to improve denoising and detection of non-FBG sensing via artificial intelligence approaches. This may be applied to improve qualitative parameters (such as resolution and accuracy) of the measurement equipment. Similarly, Brillouin optical time domain reflectometry [15,16] and optical frequency domain reflectometry [17,18] methods allow for improved signal clarity, better measurement accuracy, and enhanced signal resolution. The essence of these methods consists of processing the specified number of averaged data points in time through the data set while taking the set number of data points in the time window. This is used in various measurement techniques and is gaining attraction in distributed sensing applications.

Our focus has been on a quasi-distributed system based on FBG sensing in optical fiber, where methods based on similar (but not the same) principles can be used [19]. The goal has been to avoid complex mathematical procedures, fingerprinting databases, or artificial intelligence-based techniques [19,20,21]. In [22,23], we reported on a denoising technique based on a digital sliding window. A statistical detector was introduced to detect the spectral power of FBGs in an additive mixture of the signal and background noise. The statistical detector controls the power level depending on the given threshold level *τ*, as shown in Figure 1, indicating that the power level above the threshold belongs to the FBG signal. As is typical for FBG sensing, the statistical probability density function (pdf) achieves higher values for the FBG power spectra compared to the pdf of the background noise. This is because most of the measured values of the reflected FBG power signal are higher than the background noise values.

Detection of FBG power spectra using threshold *τ* in the overlapping power zone (where the additive mixture contains both FBG signals and background noise) brings some risk of either loss (Zone III) of FBG detection or false alarms (Zone II) due to the noise detection. Signals detected above the threshold *τ* and simultaneously above the background noise (Zone I) indicate the correct decision about the presence of the FBG. Finally, the presence of FBG below the threshold (Zone IV) is also evaluated as the correct decision if it originated from background noise, called the “rejecting detection hypothesis”. This is depicted in Figure 2. It is noted that the calculation of the detection threshold *τ* is based on Bayesian decision theory [24,25,26]. Typically, practical applications often aim to either maximize correct detection in Zone I or minimize false alarms in Zone II. Zone III in Figure 2 represents the detection loss and also shows the “significance level of the hypothesis test”. Zone IV of the correct rejection of the hypothesis also shows the “power of the test”. For example, if the desired *p_FA_* = 10^−3^, the power of the hypothesis test is sufficiently high at 1 − *p_FA_* = 0.999. It is understood that numerical values from Zones III and IV indicate whether or not statistical sampling is representative.

Neyman and Pearson showed [27] that the likelihood ratio test will maximize the power of the test for a fixed population sampling for a given false alarm probability. Therefore, the likelihood ratio test is statistically the strongest hypothesis test in the sampling signal detection theory. Then, any monotonic likelihood ratio function (i.e., pdf of FBG power spectra vs. pdf of background noise) can be used as the decision variable based on comparing mean values *μ* (of the raw observations across the entire sample) against the threshold *τ*. As a consequence, optimal decisions without prior knowledge of likelihood functions can be constructed regardless of the prior knowledge of the Gaussian process regression method (or similar) [28].

In a previous study [23], a fixed *K*-number of the discrete power spectral samples was processed using the sliding window technique where the *K*-number corresponded to a number of discrete wavelength steps within the FBG bandwidth. As was shown, *K* can be smaller. This depends on the requirement to either increase the threshold stability or to reduce the computational complexity. Statistical tests of reliability and validity showed the limits on the smallest *K* in the population sampling [24,25,26,27,28,29].

In this article, we focus on determining the least necessary but sufficient number of significant samples in statistically small population sampling while minimizing the impact on statistical numerical characteristics.

First, we investigate the impact of two-sided sliding window sampling around the cell under the test;Second, we investigate the impact of a one-sided sliding window.

In both these methods, *K* discrete steps related to the bandwidth of FBGs are applied.

Next, *K* will be gradually reduced to the smallest population sampling. This population sampling reduction will be conducted with respect to minimizing the impact on statistical detection of the FBG power spectra.

In experimental demonstrations, we will investigate how reducing *K* in the sliding window will impact the SNR and detection.

## 2. Statistical Thresholding Using Two-Sided Small Population Sampling

In this section, we study and analyze the symmetrical *K*-size two-sided population sampling, composed of left and right sub-windows, see Figure 3.

Figure 3 illustrates the main principle of statistical detection based on a sliding window. As we have already explained, the comparator in the statistical detector decides whether the power level in the cell under the test (CUT) is above the threshold level *τ*. After the presence of FBG power spectra peaks was evaluated, the window was shifted by one wavelength step, and the adjacent cell became the CUT. This is why the window is called a sliding window along the waveband. The waveband comprises *N* number of cells each containing different power levels of the signal with noise.

Based on Bayesian decision theory and the Neyman–Pearson approach, as well as the minimum required false detection, *p_FA_*, the statistical detector is not allowed to exceed the preset value of *p_FA_*. In other words, at *p_FA_* = 10^−3^, a maximum of 1 false threshold detection is allowed from 1000 CUTs investigated.

### 2.1. Statistical Threshold Calculation

The calculation of the statistical threshold *τ* uses statistical characteristics of the additive mixture of the signal and background noise. This comprises the mean *μ_K_* and the standard deviation *σ_K_*. Both characteristics are calculated from the fixed number of *K* cells from the left and the right sub-windows. This can be called *K*-sized population sampling around the CUT. By default, a symmetric *K*-size window is chosen due to the typical symmetric Gaussian shape of the reflected FBG power spectra peaks [8,9,10,11]. The size of *K* depends on the properties of FBGs, including the FBG bandwidth *B_FBG_*, sensing interrogator resolution *δ_sens_*, and the effects of attenuation. In the following example, let’s assume *B_FBG_* = 0.8 nm and *δ_sens_* = 0.008 nm. Thus, the above-threshold power can be approximated from *M* discrete wavelength steps as *M* ≅ *B_FBG_*/*δ_sens_* = 100. As a rule, it is recommended to keep *K* ≅ *M*.

First, the calculated threshold *τ* has to contain the noise function *f_SMF_* describing the attenuation approximation of the single-mode fiber (SMF-28). Second, an instrumental error function *ε_instr_* should be included. The *ε_instr_* is a sum of all instrumentation errors and includes fluctuations in the wavelength discretization, quantization, deviations, or offsets due to internal or external environmental changes. Procedures for calculating total instrumentation error are explained in [30] using qualitative parameters of involved devices in the given experimental setup. In this case, the FBG sensing instrumentation error is influenced by wavelength measurements with an accuracy of ±10 pm and power measurements of ±2%, regardless of whether a slow or fast scanning mode was used. Generally, both *f_SMF_* and *ε_instr_* functions are stabilized during long-term use, assuming stable operating conditions for the optical fiber and FBG sensing interrogator.

Next, the calculation of *τ* has to include the required *p_FA_* (for example, *p_FA_* = ×10^−3^ … ×10^−6^). The smaller the *p_FA_*, the higher values of *τ* can be achieved. However, the threshold values should range from the minimum value slightly above the background noise energy *E_min_* up to the maximum expected value of the additive mixture *E_max_* (the FBG power spectra peak value with background noise). Due to the above, the *p_FA_* is parameterized in the range from *E_min_* to *E_max_*. For large sampling populations, a full parametrization is typically required and is equal to 1. However, for decreasing *K*-size, the *p_FA_* is parameterized with lower weights. The condition *K < M* allows for the adequate weakening of the *p_FA_* parametrization.

Finally, the calculated threshold *τ* has to include the additive mixture (*N*_0_ + *E_S_*)*_k_* obtained in the given *k^th^* CUT. This value should be weighted by both the mean *μ_K_* and the standard deviation *σ_K_*. Both are obtained from the *K*-size population sampling within the sliding window. If the *K*-size is reduced, the accuracy deteriorates and the parametrization of the (*N*_0_ + *E_S_*)*_k_* value is weakened accordingly. On the contrary, the increased standard deviation increases the contribution of the additive mixture in the calculation of *τ*.

To conclude, the parameterized *p_FA_* and (*N*_0_ + *E_S_*)*_k_*, will affect the fast dynamic adaptation of the threshold *τ*. The calculation of the threshold *τ* is given by Equation (1):(1)$$\tau =\varepsilon_{instr}+f_{SMF}+\frac{-E_{max}-E_{min}}{ln\frac{1}{p_{FA}}}\frac{K-G}{K}+{\left(N_{0}+E_{S}\right)}_{k}\mu_{K}\sigma_{K}\;\;\;\left(dB\right),$$
where *G* is the number of guard cells in the neighborhood of the CUT that do not participate in the threshold calculation. In general, the more guard cells, the smaller the weight of the parameterized *p_FA_*.

### 2.2. Experimental Demonstration and Results

The experimental setup is shown in Figure 4. Figure 4a shows the investigation of FBG power spectra in a reflection mode by an optical interrogator, and Figure 4b shows a transmission mode by a stand-alone detector/optical spectrum analyzer.

The non-linear attenuation of the used optical fiber (G.652. D SMF) and the creation of the approximate broadband *f_SMF_* attenuation function is described in [31]. In the experimental demonstration, various optical fiber lengths are considered, representing the range of attenuation between −1 … −45 dB. Several FBG optical sensors with a bandwidth of *B_FBG_* ≅ 0.8 nm and maximum attenuation of −20 dB at the resonant wavelength *λ_FBG_* are connected to optical fiber.

The digitized additive mixture of the signal and background noise (in the spectral domain) is continuously processed in the predefined wavelength sliding window of different *K* sizes. This sliding window systematically shifts and *μ_K_*, *σ_K_*, and *τ* are dynamically calculated (Equation (1)) for each of the *CUT_k_*, see Figure 3. The *G* neighboring guard cells are excluded from *μ_K_*, *σ_K_*, and *τ* computing.

In Section 2.2.1 and Section 2.2.2, different threshold values of *τ* will be determined and investigated for different values of *p_FA_* for different FBG power spectra peaks and the presence of the background noise.

The interrogator processes discrete power values for each discrete wavelength in the presence of the quantization noise. To compare different effects of instrumental distortion, a commercial interrogator and a table-top analyzer with a different wavelength resolution are used in experimental investigations.

#### 2.2.1. Experimental Investigation of Two-Sided Small Population Sampling Using Interrogator

The commercial interrogator Sylex S-line S-400 [32] was used in this study, having the wavelength resolution of *δ_sens_* = 0.08 nm. Because of slightly changing BFBG under the influence of fluctuating noise, the sampling with *M* = 9 … 13 discrete values and *B_FBG_* ≅ 0.8 nm was selected. Four FBG sensors (A, B, C, and D) were deployed within the C-band. The experimental results for various detection thresholds are shown in Figure 5. In addition, interfering spectra with ten times narrower FBG bandwidths (I to X) were implemented to demonstrate the advantage of dynamic threshold adaptation. The variety of detection conditions due to partial overlap of FBG power spectra, and their varying density distribution were investigated. The calculation of threshold *τ* follows the procedure as described in Section 2.1 and includes all the listed components. It respects the schematic diagram shown in Figure 3. Sliding windows of size *K* = {8, 10, 12, 16, 24, 32, 40, 60} scan individual cells (symmetrically on left and right). This enables a step-by-step calculation of variables *μ_K_* and *σ_K_* within the C-band. Obtained results are shown in Figure 5.

Due to a rapidly rising or descending *σ_K_* at FBG power spectra peaks edges, the dynamic threshold “shakes”, especially when *K* = 8. In this case, the method is not appropriate for denoising threshold detection. However, the results for *K* = 10 or *K* = 12 indicate already adapted threshold *τ* to the additive mixture of signals and fluctuated background noise (see Equation (1)). Despite “shaky” thresholds also being seen here, they are ~0.5 to 1.5 dB, respectively, above the background noise, and therefore, the statistical detection of FBG power spectra peak levels becomes more reliable. A further increase in *K* over *M* results in “shakeless” and increased *τ* values, thus yielding a safer detection of FBG power spectra. Therefore, the recommended setting is *K* ≅ *M*.

#### 2.2.2. Experimental Investigation of Two-Sided Small Population Sampling Using Table-Top Analyzer

The analyzer AQ6370C [33] is used to process the transmitted power spectra with an oversampled wavelength resolution of *δ_sens_* = 0.0035 nm. Here, the sampling is conducted with discrete values of *M* = 90 … 120 and *B_FBG_* ≅ 0.8 nm. Within the C-band, four FBG sensors (A, B, C, and D) in the C-band are used. The results of various detection threshold scenarios are shown in Figure 6. Here, the attenuation of the optical fiber was assumed −35 dB with the highly fluctuated background noise (*σ_K_*_* N*0_ ≅ 4.3 dB) and input power of *SNR_in_* ≅ 8.5 dB. Despite these unfavorable conditions, the detectability and adaptability of *τ* are improved, compared to the previous case study.

As in the previous case study described in Section 2.2.1, thresholds for *τ* are also “shaky” for the same reasons. However, for *M* = 90 … 120 and *K* = 8 … 60, the results obtained are significantly better despite those unfavorable detection conditions. Surprisingly, even for *K* = 12 or *K* = 16, the threshold detection results are acceptable and are comparable to the previous results in Section 2.2.1 for *K* ≅ *M*.

In Figure 6, a sudden/significant drop in the threshold values *τ* in the close proximity of the FBG power spectra peaks can be noted. The deeper the drop of threshold values (especially when *K* is much less than *M*), the higher the difference, which helps to improve the SNR. These value differences are illustrated in Figure 7.

In Figure 7a, *p_FA_* ≈ 10^−3^ and *K* = 10 … 32, a random low-level false detection occurred for power levels below 1.5 dB and the detected FBG power level spectra approaching 5 dB.

When *p_FA_* ≈ 10^−4^ (case Figure 7b), all thresholds rise to their higher level. As a consequence, no false detections were observed for any *K*-size. This maintained the reliable detection of the FBG power spectra peak levels without false detections. However, the highest level of *K* = 60 (the strictest threshold) causes a decrease in those values above the threshold.

### 2.3. Threshold Behavior Analysis and Discussion

In this subsection, a mathematical analysis of the threshold calculation is presented and implications for the detection of the FBG power spectra peaks are derived.

Let us first analyze the parameterization of the 3rd component of Equation (1). We assume a typical FBG power attenuation ranging in the interval (*E_max_* … *E_min_*) = 20 dB and a typical maximum value of *p_FA_* ≤ 10^−3^. As a result, the 3rd component in Equation (1) ranges from −1.45 to −2.8 dB for the population sampling *K* = 4 … 60, and assuming 1 guard cell adjacent to the CUT in each of the sub-windows:(2)$$\frac{-E_{max}-E_{min}}{ln\frac{1}{p_{FA}}}\frac{K-G}{K}=\frac{\left|E_{max}-E_{min}\right|}{ln\frac{1}{p_{FA}}}\frac{K-G}{K}=\frac{20}{ln\frac{1}{0.001}}\frac{K-G}{K}=\;\;\;\;\;\;\;=\frac{20}{-6.90776}\frac{K-G}{K}=-2.895\frac{K-G}{K}=-2.895\left\{\begin{array}{c} \frac{4-2}{4}=0.5\\ \vdots \\ \frac{60-2}{60}=0.9\bar{6} \end{array}=\right.\;\left\{\begin{array}{c} -1.4476\\ \vdots \\ -2.7985 \end{array}\right.\;\;\left(dB\right).$$
If *K* = 60, the 3rd component parametrization is equal to −2.8 and will cause a reduced threshold *τ*. On the contrary, for *K* = 4, the 3rd component parametrization is equal to −1.45 and will cause an increased *τ*, see Equation (1). This property can be used to set the value of *τ* which will be used later.

Next, the 4th component in Equation (1) will be analyzed in the presence of background noise only ((*N*_0_ + *E_S_*)*_k_* = *N*_0*k*_). Here, the *K* value affects the threshold calculation through changes of statistical characteristics of *μ_K_* and *σ_K_* as follows:(3)$$\mu_{K}=\frac{1}{K}\sum_{i=1}^{K}x_{i}=N_{0}=\sum_{i=1}^{K}x_{i}\left\{\begin{array}{c} \frac{1}{30-2}=\frac{1}{28}=0.0357\\ \vdots \\ \frac{1}{60-2}=\frac{1}{58}=0.0172 \end{array}\right.\;\;\left(dB\right)\;,$$
(4)$$\sigma_{K}=\sqrt{\frac{\sum_{i=1}^{K}{\left(x_{i}-\mu_{K}\right)}^{2}}{K-1}}=\frac{\sqrt{\sum_{i=1}^{K}{\left(x_{i}-\mu_{K}\right)}^{2}}}{\sqrt{K-1}}=\sqrt{\sum_{i=1}^{K}{\left(x_{i}-\mu_{K}\right)}^{2}}\left\{\begin{array}{c} \frac{1}{\sqrt{30-2}}=0.189\\ \vdots \\ \frac{1}{\sqrt{60-2}}=0.131 \end{array}\right.\;\left(dB\right)\;.$$
Finally, the total contribution of the 4th component to the threshold calculation of Equation (1) without the occurrence of FBG power spectra peak is:(5)$$\mu_{K}\sigma_{K}=\sum_{i=1}^{K}x_{i}\sqrt{\sum_{i=1}^{K}{\left(x_{i}-\mu_{K}\right)}^{2}}\left\{\begin{array}{c} 0.0357\cdot 0.189=0.006747\\ \vdots \\ 0.0172\cdot 0.131=0.002253 \end{array}\right.\;\left(dB\right).$$
From the above, it can be shown that an approximately 3-fold increase in the contribution of the 4th component (from 0.002253 in the case of *K* = 60 to 0.006747 in the case of *K* = 30, respectively) can be achieved, thus contributing to the threshold level increases.

Let’s now analyze the *μ_K_*-parameterization of (*N*_0_ + *E_S_*)*_k_* in Equation (1) when the approximately Gaussian-shaped FBG power spectra peak overlaps with the sliding window. Using the example from Section 2.2.1, and using *K* = *M*/2 ≅ 60, the mean values keep increasing from the lowest to the highest values just in 30 wavelength steps. At the 60th step, where *CUT_k_* contains the FBG power spectra peak maxima, the two-sided sampling reaches the value *μ*_*K*=60_ ≅ 2 ∙ 0.25 ∙ (*E_max_* … *E_min_*) above the *N*_0_ noise level, see Figure 8a. When the sliding window touches the falling edge of the FBG power spectra peak and starts to leave it, the mean value *μ_K_* starts decreasing. However, for *M*/2 ≅ 60 and *K* = 30, the behavior of the mean values will remain mostly unchanged. To be noted, for the 60th step, the shorter the sliding window, the higher the mean value *μ_K_*. When *K* = 10, *μ*_*K*=10_ ≅ 0.707 ∙ (*E_max_* … *E_min_*).

Now we analyze the *σ_K_*-parameterization of (*N*_0_ + *E_S_*)*_k_* in Equation (1) when the approximately Gaussian shaped FBG power spectra peak overlaps with the sliding window. This is shown in Figure 8b. For *K* = *M*/2 ≅ 60, the standard deviation values achieve the highest values in the ~30th and ~90th steps. Here, the square root multiplier in Equation (4) achieves the widest span of input values. It is worth noting that the *σ_K_*-parametrization on the leading and falling edges can reach similar effects as the *μ_K_*-parametrization. This depends on the steepness of the edges. When the sampling window slides from the 30th to the 90th step, the *σ_K_* value drops. For *M* ≅ 60 and *K* = 30, the behavior of the standard deviation values will be similar to the case of *K* = *M*/2 ≅ 60. To be noted, the longer the sliding window, the larger the standard deviation *σ_K_*. This is the origin of threshold adaptability.

Finally, a comparison of the magnitudes *μ_K_* and *σ_K_* in Figure 8 shows the difficulty of meeting the three-sigma rule (known also as the 68-95-99.7 rule) that is used to verify the normality of population sampling. This rule is also used for percentage quantification of reliability of population sampling from selected values (here, the selection of cells in the two-sided sliding window) in the following way. If ~68% of these values are from the interval (*μ* ± 1*σ*), ~95% from the interval (*μ* ± 2*σ*), and 99.7% from the interval (*μ* ± 3*σ*), respectively, a randomly selected sample can be considered the Gaussian normal distribution. As an example, we analyze one of the measured FBG spectral peaks measured by the table-top analyzer (see Section 2.2.2). Here, the FBG B maxima of *μ_K_* values spanning from −70.3659 to −69.3485 dB (depending on *K*-size) is reached for *λ_FBG_* = 1547.25 nm (see Figure 7a). This corresponds to values of *σ_K_* spanning from 0.08286 to 1.01195 dB (see Figure 7b). Based on the three-sigma rule, 99.7% of the values should have been within the interval (−69.3485 ± 3 ∙ 0.08286) = (−69.5971 … −69.0999) dB for two-sided *K* = 10 but is not. As shown in Figure 6, the values inside the sliding window are from the interval (−69.308 … −68.734) dB. Similarly, the three-sigma rule is not fulfilled for two-sided *K* = 60 because the values span in interval (−78.688 … −68.734) dB, which is out of the required interval (−70.3959 ± 3 ∙ 1.01195) = (−73.43175 … −67.36) dB, see Figure 6. The same applies to the other *K*-sizes and *μ* and *σ* values of other wavelengths. However, it needs to be noted that in cases for *K* = 10 … 16, the three-sigma rule is less broken compared to cases for *K* = 40 … 60. In spite of this, the use of two-sided small population sampling is reliable for successful statistical threshold detection. This is illustrated by results in Figure 5, Figure 6 and Figure 7.

## 3. Statistical Thresholding Using One-Sided Small Population Sampling

In this section, we study and analyze the impact of the population sampling using one (left) sided window having an asymmetrical *K/2*-size, see Figure 9.

### 3.1. Statistical Threshold Calculation

The calculation of the statistical threshold *τ* uses the mean *μ_K/_*_2_ values and the standard deviation *σ_K/_*_2_ values obtained from the fixed number of *K/2*-cells contained in the left sub-window. Please note that the sliding window is asymmetrically located, in this case sitting on the left side, see Figure 9. This reduces the computation complexity by excluding the right-side sub-window from population sampling. In the next step, we will investigate the impact of this approach on the quality of the threshold detection results. Since the shape of the reflected FBG power spectra is typically a symmetric Gaussian function, we need to learn if in this approach the *μ_K/_*_2_, *σ_K/_*_2_, and *τ* would differ from the two-sided *μ_K_* and *σ_K/_*_2_, respectively. Similar to Section 2.1, the calculation of the threshold *τ* will use Equation (1).

### 3.2. Experimental Demonstration and Results

To compare the effect of halving the population sampling, the same considerations, instrumentation, and conditions are applied in the experimental investigation as in the previous Section 2.2.

#### 3.2.1. Experimental Investigation of One-Sided Small Population Sampling Using Interrogator

Here, as described in Section 2.2.1, the same commercial interrogator and the same deployment scenario of four FBG sensors A, B, C, and D along the C-band with *M* = 9 … 13, interfered by narrowband FBGs I … X, was used.

Results are shown in Figure 10 indicating various threshold detection. As the sliding window approaches individual FBG power spectra levels (Note: sliding window shifts from lower to higher wavelengths), the thresholds keep increasing very slowly. This is due to a weak mean value of *μ_K/_*_2_. Please compare the results in Figure 5 and the discussed effect of *μ_K_* in Section 2.3. After passing through the FBG power spectra peaks, the threshold behavior stabilizes. This is similar to the behavior shown in Figure 5. However, the cell reduction from *K* to *K/2* noticeably causes higher fluctuations of *σ_K/_*_2_ thus calculated *τ*, especially when *K/2* < *M*. Therefore, this scenario is not recommended for system operation. On the other hand, thresholds *τ* for *K/2* = 8 … 16 ≅ *M* adapt very well to the signal/noise behavior. For further increases in *K/2*, when *K/2* > *M*, threshold *τ* rises accordingly, which may lead to the power loss related to the right side of the FBG power spectra. In summary of the above, it is recommended to use *K/2* ≅ *M*.

#### 3.2.2. Experimental Investigation of One-Sided Small Population Sampling Using Table-Top Analyzer

The same table-top analyzer was used to process the transmitted power spectra of four deployed FBG sensors within the optical fiber C-band using values *M* = 90 … 120. The experimental results of various detection thresholds *τ* are shown in Figure 11.

It can be noted again that the thresholds “shake”, here slightly more than in the case illustrated in Section 2.2.2. This is due to the smaller mean *μ_K/_*_2_ values compared to “shaky” *μ_K_* values. As the sliding window approaches FBG power spectra peaks, the threshold values increase slowly and better adapt to FBG power spectra levels with background noise. After passing the FBG power spectra peak maxima, the behavior of the threshold levels stabilizes similarly to those in Figure 6. A greater “shaking” was observed for *K/2* = 4 … 6 << *M*. This causes rising in false detection. For *K/2* = 8 … 16 (smaller than *M*), the detection results are acceptable and comparable to the investigation in Section 2.2.2 with *K* = 12 or *K* = 16 when *K* = *M*. The previously observed phenomenon of dropping *τ* values in the vicinity of FBG power spectra peaks appeared also here and again helped to improve the SNR, see Figure 12.

The detected power levels of the 4 FBGs by using one-sided sliding windows for *K/2* = 5 … 30 are shown in Figure 12. In cases when *K/2* = 5 or *K/2* = 6, thresholds are “shaky” thus the threshold detection is not reliable, leading to increased false detections. In contrast to Figure 7, all FBG power spectra are tilted and sharpened. This is an artifact caused by one-sided population sampling. When *p_FA_ ≈* 10^−3^ (see Figure 12a), false detections are noted at levels below 1 dB when *K/2* = 8. No false detections occur for *K/2* > 8. Contrary to the results shown in Figure 7, here the detected power levels of the FBG power spectra are slightly higher, thus the one-sided population sampling performs better than using two-sided population sampling in Section 2.2.2.

When *p_FA_ ≈* 10^−4^, shown in Figure 12b, all threshold levels are increased. Therefore, no false detections are noted when *K/2* = 8 … 30. Here, the large fluctuation of the background noise resulted in the “shaky” threshold behavior. Similar to the situation in Figure 12a, the *K/2* = 12 … 30 values allow maintaining the FBG power spectra levels at the reliable detection level thus without false detections. However, in contrast to Figure 7, there is no loss in the FBG power spectra detection when *K/2* = 60.

### 3.3. Threshold Behavior Analysis and Discussion

In this subsection, a brief analysis of the threshold calculation and behavior is given.

Let’s analyze the parameterization of the 3rd component of Equation (1). Considering the same conditions as in Section 2.3, but half *K* to *K/2* = 2 … 30, Equation (2) will be as follows:(6)$$\frac{-E_{max}-E_{min}}{ln\frac{1}{p_{FA}}}\frac{K/2-G}{K/2}=\frac{20}{ln\frac{1}{0.001}}\frac{K/2-G}{K/2}=2.895\left\{\begin{array}{c} \frac{2-1}{2}=0.5\\ \vdots \\ \frac{30-1}{30}=0.9\bar{6} \end{array}=\right.\;\left\{\begin{array}{c} -1.4476\\ \vdots \\ -2.7985 \end{array}\right.\;\;\left(dB\right).$$
The solutions of Equations (2) and (6) in terms of *K* is a value equal to *K/2* that is identical for both one-sided and two-sided population sampling.

Let us now analyze the parameterization of the last component of Equation (1) with the presence of the background noise only, (*N*_0_ + *E_S_*)*_k_* = *N*_0*k*_. Here, the *K/2* value affects the threshold calculation through changes of statistical characteristics of the *μ_K/_*_2_ and *σ_K/_*_2_:(7)$$\mu_{K/2}=\frac{1}{K/2}\sum_{1=1}^{K/2}x_{i}=N_{0}=\sum_{1=1}^{K/2}x_{i}\left\{\begin{array}{c} \frac{1}{15-1}=\frac{1}{14}=0.0714\\ \vdots \\ \frac{1}{30-1}=\frac{1}{29}=0.0344 \end{array}\right.\;\;\left(dB\right)\;,$$
(8)$$\sigma_{K/2}=\sqrt{\frac{\sum_{i=1}^{K/2}{\left(x_{i}-\mu_{K/2}\right)}^{2}}{K/2-1}}=\sqrt{\sum_{i=1}^{K/2}{\left(x_{i}-\mu_{K}\right)}^{2}}\left\{\begin{array}{c} \frac{1}{\sqrt{15-1}}=0.267\\ \vdots \\ \frac{1}{\sqrt{30-1}}=0.186 \end{array}\right.\;\left(dB\right)\;$$
Here, the numerical values of *μ_K/_*_2_ found in Equation (7) are half of the *μ_K_* values found in Equation (3). Because the number of the selected values *x_i_* is also halved, then *μ_K/_*_2_ ≅ *μ_K_* in cases of uniform statistical distribution. The numerical values of *σ_K/_*_2_ found from Equation (7) differ from *σ_K_* given by Equation (4). In summary, due to computational demands, the selection of *K/2* over *K* is preferable but is governed by the availability of a number of cells with required properties.

Based on Equation (7), we have also analyzed the *μ_K/_*_2_-parameterization of (*N_0_* + *E_S_*)*_k_* in the case of when the leading edge of FBG power spectra overlaps with the sliding window. This is shown in Figure 13a in the case of small population sampling when *K/2* = 4 … 30. The increase of the mean *μ_K/_*_2_ values is similar to the case of *μ_K_* when *K* = 8 … 32 (see Figure 8a). Overall, the behavior of the mean of *μ_K/_*_2_ values related to any FBG power spectra in the presence of noise is nearly identical, except when *K* = 32 … 60 where *μ_K_* is slightly lower, see Figure 8, and *σ_K_* fluctuates massively.

Based on Equation (8), we then analyzed the *σ_K_*_/2_-parameterization of (*N*_0_ + *E_S_*)*_k_* in the case when the leading and falling edges of FBG power spectra are part of the sliding window. For *K/2 << M* or *K/2 < M*, the standard deviation exhibits two local maxima in close proximity to the FBG power spectra peak, see Figure 13b. It can be seen that the *σ_K/_*_2_ value drops in between the two *σ_K/_*_2_ maxima. The smaller the population sampling, the deeper the drop of *σ_K/_*_2_. When comparing *σ_K/_*_2_ in Figure 13b to *σ_K_* in Figure 8b, due to smaller population sampling, the *σ_K/_*_2_ maxima values are smaller. The left one-sided population sampling causes a higher rise of the left *σ_K/_*_2_ maxima compared to the one on its right. This obscures the threshold detection levels. From the above and Figure 10 and Figure 11, it can be concluded that the thresholds on the left side of the FBG power spectra peaks are better adapted to the signal plus background noise levels. Therefore, the outcomes of the “*K/2* approach” described in Section 3 are superior to those “*K* approach” described in Section 2.

Finally, a comparison of the magnitudes of *μ_K_* and *σ_K_* in Figure 13 indicates that the three-sigma rule has not been met. Figure 14 reflects an in-depth analysis of the non-fulfillment of the three-sigma rule. It is based on 8750 measurement points taken within the optical fiber C-band. The results obtained for different sizes of *K* from 4 to 60 are shown for two-sided (blue line) and one-sided (magenta line) sliding windows, respectively. It can be seen that the three-sigma rule is better fulfilled for sliding window *K* = 11, …, 16 (when compared to the rest of the cases). Near plateau response has been observed for both plotted dependencies when *K* was between 22 and 60 leading to 45 to 56 cases of the three-sigma rule violations. The local minima of both dependencies can be seen around *K* = 12 (11 to 16 is acceptable). In a given additive mixture of the FBG power spectra signal and background noise, those values are considered as optimal and, therefore, recommended for denoising and threshold detection. It can also be concluded that, despite the fact that the three-sigma rule is *not* fulfilled, the detection using one-sided small population sampling would also be reliable. This can be confirmed by examining Figure 5, Figure 6, Figure 7, Figure 10, Figure 11 and Figure 12.

## 4. Conclusions

In our recent [23] digital sliding window denoising technique, a number of discrete power spectral population samples were processed. In order to increase computational efficiency, it is possible in some cases to reduce population sampling while maintaining the success of statistical detection. In this article, we focused on determining the small population sampling for sufficient detection of fiber Bragg gratings power spectra in an optical fiber sensing system. For such statistical threshold detection, the highest allowed number of false detections is set, which is based on the Bayesian principle.

In this article, the two-sided and one-sided statistical detectors have been introduced with reduced population sampling. In addition to the explanation of the method and the introduction of the main algorithms, a mathematical assessment of the impact of statistical characteristics, mean, and standard deviation, are presented for various population sampling reductions. Next, reduced population sampling is applied using two common instrumentations for fiber optic sensing: a commercial interrogator with standard wavelength resolution and a laboratory analyzer with improved wavelength resolution. We thereby confirmed the success of the statistical threshold detection under various conditions of fluctuating background noise, signal-to-noise ratio, approaching the adjacent fiber Bragg grating power spectra, and interferences by other signals. As from the demonstrated examples, statistical characteristics’ impact on statistical threshold detection was deeply analyzed for different false detection requirements. We have also shown that for the two-sided *K* = 11 … 16 and for the one-sided *K/2* = 5 … 8 population sampling the majority of cases obey the three-sigma rule. As a result, in the case of a reduced number of samples (11 to 16), the denoising and detection will benefit from implementing the two-sided sliding window. Similarly, in the case of implementing the one-sided sliding window, using 5 to 12 samples is recommended. For higher *K* values, where the three-sigma rule is only loosely fulfilled, some decrease in the detected FBG power spectra will be observed.

## Figures and Tables

**Figure 1 sensors-24-02285-f001:**
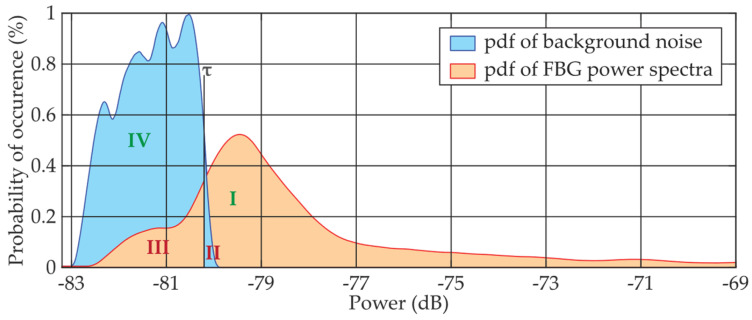
Principle of statistical detection of the FBG power spectra peaks in the additive mixture of the signal and background noise, where pdf means the respected probability density function, *τ* is the threshold in dB and I–IV are zones of correct or incorrect detection determined by the *τ* and pdfs.

**Figure 2 sensors-24-02285-f002:**
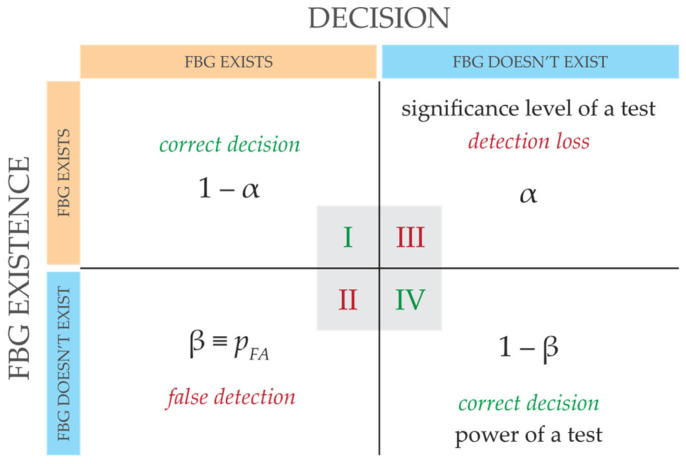
Basic principle of the decision-making process, where I–IV are zones of correct or incorrect detection determined by the threshold *τ* and pdfs.

**Figure 3 sensors-24-02285-f003:**
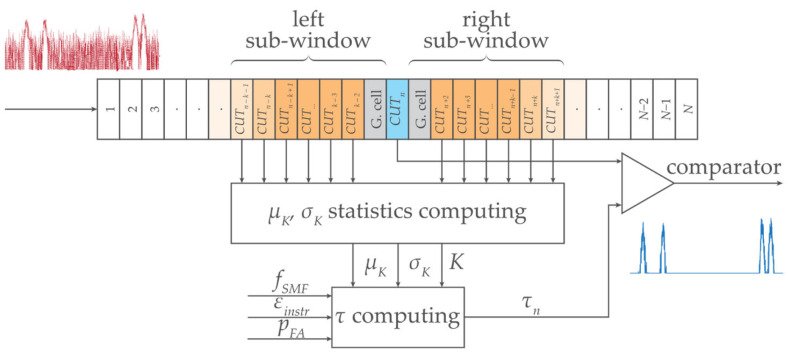
Concept of the statistical threshold detector of FBG power spectra peaks level with a symmetric two-sided *K*-size sliding window.

**Figure 4 sensors-24-02285-f004:**
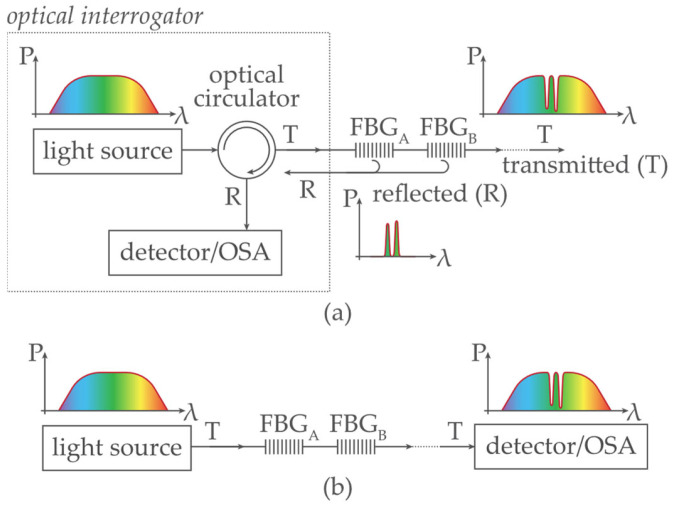
FBG sensing experimental setup using (**a**) an interrogator; (**b**) a stand-alone light source and a detector/optical spectrum analyzer.

**Figure 5 sensors-24-02285-f005:**
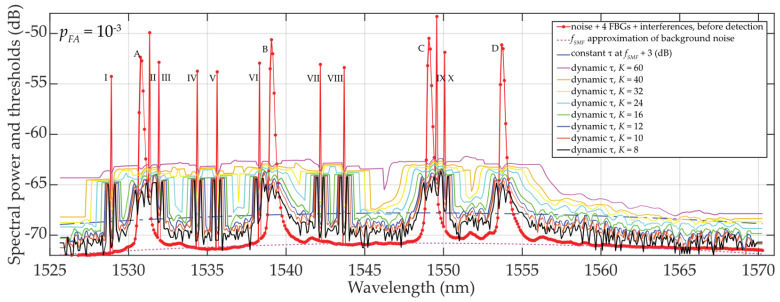
Results of dynamic statistical threshold detection of the wideband (A, …, D) and narrowband (I, …, X) FBGs with different power levels, *p_FA_* ≈ 10^−3^ and *K* = 8 … 60.

**Figure 6 sensors-24-02285-f006:**
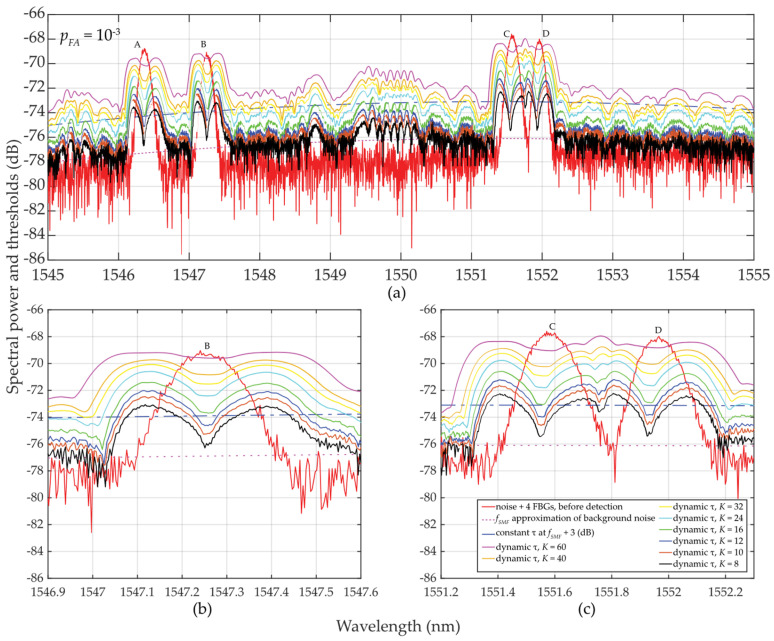
Results of dynamic statistical threshold detection of the wideband FBGs with different power levels, *p_FA_* ≈ 10^−3^ and *K* = 8 … 60: (**a**) for (A, …, D) FBGs; (**b**) detailed view for B FBG; (**c**) detailed view for C and D FBGs.

**Figure 7 sensors-24-02285-f007:**
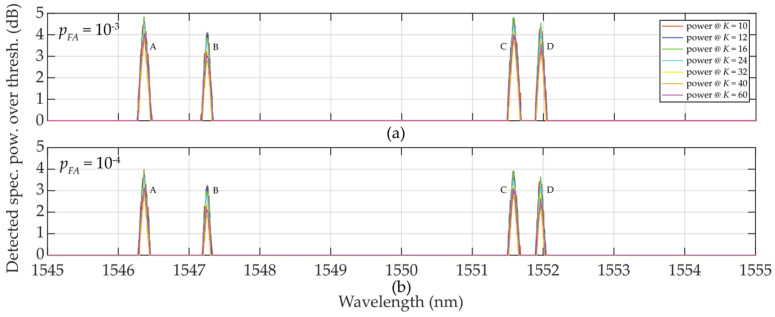
Illustration of value differences of the wideband (A, …, D) FBGs when *K* = 10 … 60: (**a**) *p_FA_* ≈ 10^−3^; (**b**) *p_FA_* ≈ 10^−4^.

**Figure 8 sensors-24-02285-f008:**
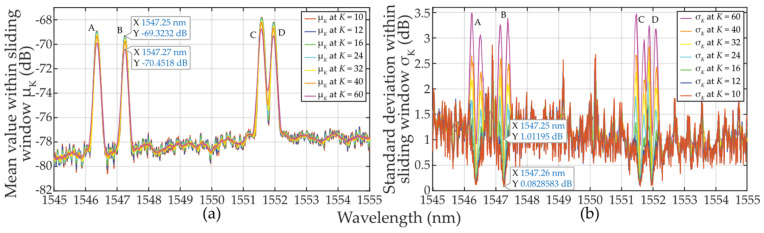
Statistical characteristics of the additive mixture of the (A, …, D) FBGs signal and background noise within sliding windows when *K* = 10 … 60: (**a**) mean values *μ_K_*; (**b**) standard deviation values *σ_K_*.

**Figure 9 sensors-24-02285-f009:**
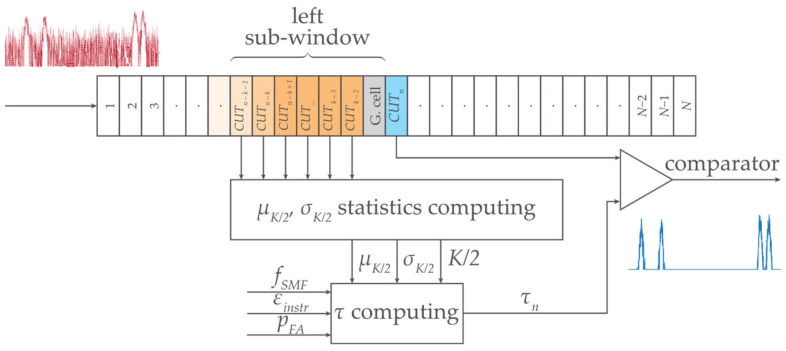
Concept of the statistical threshold detector of FBG power spectra peaks level with an asymmetric one-sided *K/2*-size sliding window.

**Figure 10 sensors-24-02285-f010:**
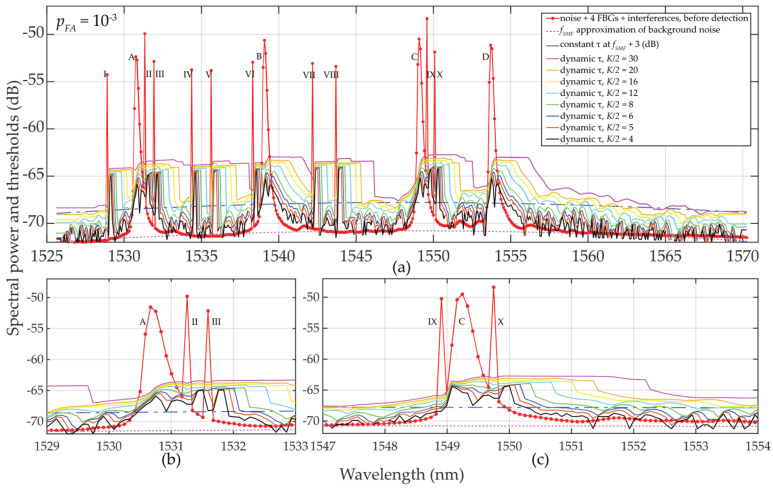
Results of dynamic statistical threshold detection (**a**) of the wideband (A, …, D) and narrowband (I, …, X) FBGs with different power levels, *p_FA_* ≈ 10^−3^ and *K/2* = 4 … 30; (**b**) detailed view for A, I and II FBGs; (**c**) detailed view for C, IX and X FBGs.

**Figure 11 sensors-24-02285-f011:**
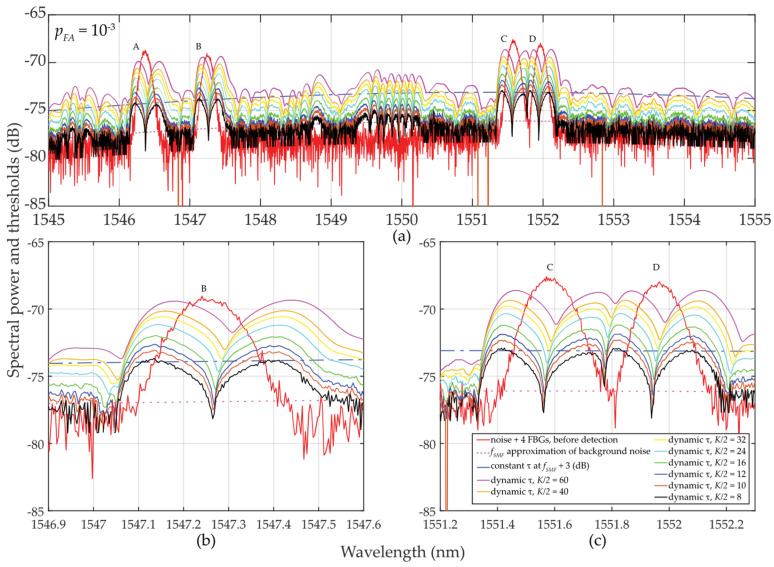
Results of dynamic statistical threshold detection of the wideband FBGs with different power levels, *p_FA_* ≈ 10^−3^ and *K/2* = 4 … 30 (**a**) for (A, …, D) FBGs; (**b**) detailed view for B FBG; (**c**) detailed view for C and D FBGs.

**Figure 12 sensors-24-02285-f012:**
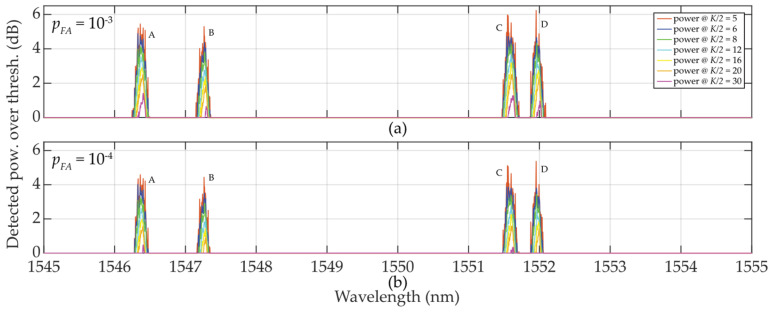
Illustration of value differences of the wideband (A, …, D) FBGs when *K/2* = 8 … 30: (**a**) *p_FA_* ≈ 10^−3^; (**b**) *p_FA_* ≈ 10^−4^.

**Figure 13 sensors-24-02285-f013:**
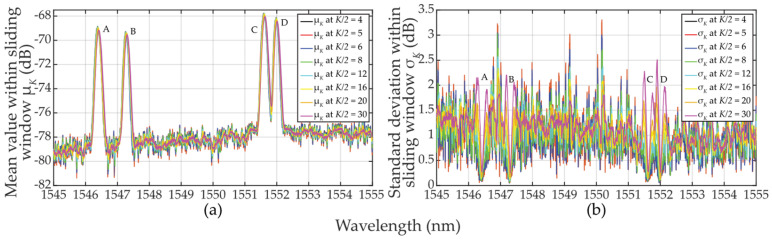
Calculated statistical characteristics of the (A, …, D) FBG power spectra levels plus background noise within sliding windows when *K/2* = 4 … 30: (**a**) mean values *μ_K_*; (**b**) standard deviation values *σ_K_*.

**Figure 14 sensors-24-02285-f014:**
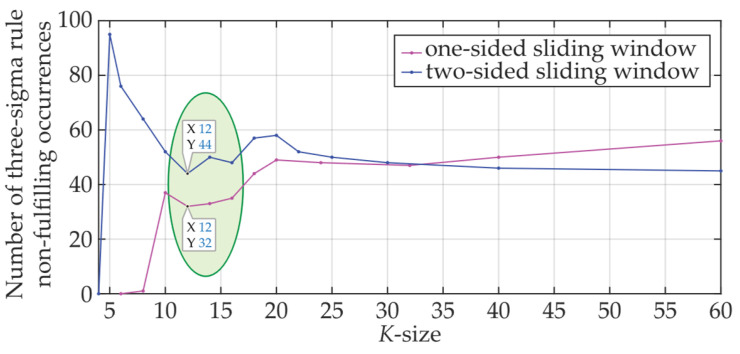
Number of three-sigma rule non-fulfilling occurrences for the mixture of the FBG power spectra and background noise for *K* = 4, …, 60 when using a two-sided (blue line)/one-sided (magenta line) sliding window, respectively.

## Data Availability

Data files or software solution can be made available after meaningful agreement with the authors.

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
