# Peer review of "Impact of Reducing Statistically Small Population Sampling on Threshold Detection in FBG Optical Sensing"

_sensors, 2024, doi:10.3390/s24072285_

Round 1

Reviewer 1 Report

Comments and Suggestions for Authors

This article focuses on determining the least necessary number of significant samples in a statistical population sampling, while minimizing the impact on the statistical detection of the fiber Bragg grating (FBG) power spectra. From results achieved, it was found that the normality three-sigma rule does not need to be followed while using the small population sampling. Experimental demonstrations and analyses showed that the proposed concept of denoising and statistical threshold detection based on the sliding window does not depend on the prior knowledge of probability distribution functions of the FBG power spectra peaks level and background noise. This manuscript is well written and clearly presented, I would suggest an acceptance of this article.

Here is some other comments, 1. The legends in Figures 5 and 10 cover a relatively large area of the figure, which does not show a full figure. 2. The last part of section does not meet the three-sigma rule,I would suggestion more threatical work.

Reviewer 2 Report

Comments and Suggestions for Authors

The article under review focuses on determining the least necessary number of significant samples in a statistical population sampling, while minimizing the impact on the statistical detection of the fiber Bragg grating (FBG) power spectra. This is achieved by using Bayesian decision making, small population statistics while excluding false detection. 

I have the following notes:

1. Full affiliations, including places of work, city and country, must be included in the work.

2. Despite the fact that the connection diagrams (experimental setups) for fiber Bragg gratings are quite simple and obvious, I would ask the authors to add a drawing for a wide range of readers.

3. It seems to me that the literature review presented in the article is incomplete. First, it is necessary to answer the question, why can’t we use an approximation similar to that used for the Lorentzian function in Brillouin analysis? In the case of BOTDA/BOTDR, the method is called Lorentzian curve fitting. Of course, you need to use a Gaussian function fitting in your case. A set of modern methods can be seen, for example, here: http://dx.doi.org/10.1134/S0020441222050268 . Further, there are modern methods for processing FBG spectra specifically. Say, like these ones: https://doi.org/10.1109/IMOC.2007.4404351 , https://doi.org/10.1016/j.measurement.2013.07.029 , https://doi.org/10.3390/s21082817 . In addition, I would suggest consider the use of the Frequency Domain Dynamic Averaging (FDDA) method, which has been successfully used to process similar spectra and increase their signal-to-noise ratio.

4. When discussing the results of the work, many numbers are given directly in the text of the article. The visualization presented in the article, in my opinion, does not provide a clear idea of the results. I propose to additionally depict them as follows: for the cases of single and double windows, draw graphs in which some accuracy criterion (accuracy, precision, wavelength error, etc) will be presented as a function of the number of samples. This way, a break in the function will be noticeable, which will clearly show the optimal value. Do you think this is possible? If so, I beleive it will improve the article.

5. Some pictures are at the end of sections, I would suggest moving them higher, right after the first mention in the text.

6. I suggest that authors pay more attention to document formatting: some sections of the article are not indented.

7. The abbreviation 'pdf' must be explained in the text before the first mention in Figure 1.

Reviewer 3 Report

Comments and Suggestions for Authors

There are many aspects that need clarification before it can proceed for publication. Below, I have outlined my comments:

Comment 1

- A significant concern regarding your abstract, it is the lack of clarity in conveying the specific objectives and contributions of your research

Comment 2

- In the introduction there is a lack of discussion on the current state-of-the-art.

Comment 3

- Figure 1 and 2 need more explanation, they are not analyzed well. Authors should give a deep explanation of this figures.

Comment 4

-Figure 2 in not clear (I zoomed 200 % to read the word FBG excites and FBG doesn’t excite ).

Comment 5

-What are the Bayesian decision theory and Neyman-Pearson criteria??

You mention them many times without any detailed explanation.

Comment 6

- In section 2.1 While the authors mention an example involving FBG bandwidth and sensing interrogator resolution, they don’t provide clear, step-by-step examples of how the statistical threshold is calculated

Comment 7

- Authors introduce parameters such as instrumental error function and required false alarm probability (pFA) without adequately discussing their assumptions and limitations.

Comment 8
- The authors in the conclusion provide recommendations for reduced population sampling in statistical threshold detection but fail to offer clear guidelines or criteria for determining the optimal number of samples.

Round 2

Reviewer 3 Report

Comments and Suggestions for Authors

I accept the paper in its actual form.